# Understanding Organisational Risks and Opportunities Associated with Implementing Australia's National Disability Insurance Scheme from the Nonprofit Service Provider Perspective—Findings from Quantitative Research

**Hamin Hamin [1], David Rosenbaum [2,*] and Elizabeth More [3]**

1    Magister Manajemen, Universitas Budi Luhur, Jakarta 12260, Indonesia
2    King Own Institute, Sydney, NSW 2000, Australia
3    Study Group, Sydney, NSW 2000, Australia
***    Correspondence: david@koi.edu.au

**Abstract:** In this paper, we provide useful lessons from a quantitative analysis across several nonprofit organisations undergoing generational change due to the implementation of the Australian government's National Disability Insurance Scheme (NDIS). This paper contributes to the field in demonstrating the usefulness of the approach in revealing how change has to occur at both the micro and macro levels of the organisations involved, affecting both followers and transforming leadership, whilst simultaneously reinforcing the need to address the strategic and operational risks inherent in such transformational change. It represents a follow-up to an earlier published longitudinal qualitative research and provides further evidence on the key findings associated with the development of the NDIS Implementation Framework. The current paper considers the importance of the risk and opportunity conundrum associated with the implementation of the NDIS among Australian nonprofit service providers. This paper recognises that, as entities operating ostensibly outside the purely commercial realms of service design and delivery, nonprofit service providers are potentially handicapped by an historic lack of relevant and necessary market-based skills. The risks necessitate an accelerated programme of skill development and skill acquisition to enable the full range of opportunities to be realised. The change management processes, identified using the conceptual framework of readiness → implementation commitment → sustainability, as discussed in this paper, highlight the potential financial consequences which have substantial impacts on such nonprofit service providers. Organisations in these settings are challenged by ongoing financial sustainability issues where very small financial margins, resulting directly from the generational business model shift from a supply-driven system to a demand-driven system, may prove the difference between organisational survival and failure.

**Keywords:** transformational change; nonprofits; NDIS Implementation Framework; business models; sustainability; financial risk

## 1. Introduction

The National Disability Insurance Scheme (NDIS) represents a generational shift in the way in which services are delivered to people with a disability. Prior to its initial roll-out, disability services were more securely funded within a supply-driven process, meaning that governments at the Federal and State levels funded service delivery organisations directly to provide a wide range of standardised services to people with a disability. The bulk of these services were provided by nonprofit service providers. The NDIS sought to restructure this process and introduce a demand-driven system whereby 'funding packages' were developed by the Federal Government. These represented a maximum annual dollar value for packages based on the complexity of medical needs of people with a disability.

With their packages confirmed, these individuals would source their required services from registered nonprofit as well as commercial service providers, meaning that funding would now flow from the government to people with a disability, who would then source their services from the market, resulting in what has become recognised as a more financially risky demand-driven system for providers. In this way, services could become more bespoke, responding to the needs of their clients.

"The instrument of collaborative inquiry . . . is an approach in which the scholar, aware of the problems that managers and operators daily experience in the field, sets up a collaborative study of these needs through engagement and active involvement of organization members and collaborative interaction with management" (Shani 2021, p. xiii). Whilst ours was a very broad canvas of collaborative learning, since we were focused on a particular nonprofit sector involved in the change management of introducing the NDIS, across several organisations, our participant organisations were involved in interviews for an earlier qualitative research (Rosenbaum and More 2022) and surveys for this quantitative research, as well as some follow-up meetings and presentations to participant organisation boards. Our aim for the qualitative research was to develop a pragmatic framework to enhance the practice of managing the major change wrought by the NDIS to diverse large and small organisations in the sector, to offer a pragmatic tool for managing such changes for both organisations and policymakers, and to add to the body of change management literature. Our aim for this quantitative research is to assess the efficacy of the National Disability Insurance Implementation Framework (NDISIF), developed in the initial qualitative study. As reflected in capturing our intent with the research: "The focus of collaborative inquiry is to generate the practical knowledge that enables the organization to make relevant changes and to contribute actionable knowledge to the social science of organizational change and development." (Shani 2021, p. 7). In doing so, we seek to aid nonprofit service delivery organisations in successfully implementing the NDIS and, in doing so, mitigate the risks of failure, while visioning and acting upon the organisational opportunities offered by the NDIS, and, in the bigger picture context, ensuring the long-term success of the NDIS for the sake of people with disabilities. This is crucial given the ongoing escalation of cost blowouts with the system.

The change wrought by the implementation of the NDIS brings with it many strategic and operational risks for employees, involving multiple dimensions of stress, uncertainty, and diverse responses of various kinds. It tests the very change capability, adaptability, organizational learning, sustainability, risk management, and flexibility of organisations and their leadership. Organisational change is impacted by context and environment, and dynamic change in a reciprocal way, that may impact the very sustainability, nature, structure, and processes of the organisation. This is as true for the nonprofit sector as for the commercial sector, although variables, constraints, and opportunities may differ.

The NDIS is a generational shift in, and an altered model of, the way services to people with disabilities are developed, provided, and funded. Its focus is on a person-centred approach in which services are reflective of individuals who remain active participants and decision makers in their own lives (Green and Mears 2014). Whilst it is not unique in a global context, it represents a major paradigm shift in Australia. It is a prime example of a force for good, given that it reflects the rights of people with a disability to live free from abuse, exploitation, and violence. These are in keeping with Australia's commitment to the United Nations' Convention on the Rights of Person with Disabilities (UN 2006), and reflected in the Vision Statement in the recent Australia's Disability Strategy 2021–2031 as " . . . an inclusive Australian society that ensures people with disability can fulfill their potential, as equal members of the community" (Commonwealth of Australia 2021, p. 2).

At the heart of the creation and implementation of the NDIS is also the context of the United Nations' Sustainability Goals, particularly those of Goal 10—Reduce Inequalities, and Goal 16—Stand Up for Human Rights. One clear challenge to fully achieving both Disability and Sustainability goals is the need for a sustainable funding model which adequately addresses the associated funding risks, which have been prophesized in the

early days of the system and are now gathering pace as the costs continue to escalate. The NDIS is not alone in this challenge as Gilchrist and Perks (2022, p. ii) state: "Indeed, many not-for-profit social services organisations are challenged in terms of sustainability by what might be termed a malevolent cycle, where poor quality impacts staffing, in turn impacting service capacity, thereby reducing income while, at the same time, infrastructure and other elements must continue to be paid for". As Gilchrist and Perks (2022, p. 5) puts it: "The survival of the Not-for-profit and/or charitable corporation is a secondary issue to the sustainability of services given the risk borne by people with disability who need reliable, appropriate quality services and supports in order to live their lives".

The introduction of the NDIS in Australia in 2016, which is yet to be fully implemented, shifts the focus of service design, delivery, and financial support from a supply-driven business model to a demand-driven business model. The former is structured around service providers developing programmes and being funded by Federal and State Governments. The demand-driven approach requires service providers to develop and deliver programmes based on the requirements and demands of service users, with the Federal Government providing "packaged" funding directly to service users who then pay service providers for using services that meet their specific needs (Rosenbaum and More 2022). These "needs" are reflective of their specific disabilities and their aspirations for living a fulfilled life (Taylor et al. 2020). The origins of the NDIS sought, in part, to address the "contracting culture" inherent in the steady shift from the state to the market, which flowed from the impact of neo-liberalism in Australia (Onyx et al. 2016). However, the political repercussions of ongoing funding for the NDIS have not been fully successful. The Federal Government has sought to increasingly contract a range of operational activities that underpin the NDIS to the private sector, moving away from the person-centric focus and negatively impacting those the NDIS originally sought to benefit.

The NDIS offers a new challenge and associated risks as well as opportunities to clients and organisations, representing a mindset and generational shift for those with disabilities being supported with services that enable them to lead fruitful and meaningful lives (Meltzer and Davy 2019). The focus of change is the movement from a supply-driven approach to a demand-driven one, with market mechanisms becoming the focus of both service design and service delivery (Foster et al. 2021). The former model relies on service providers developing programmes and being historically funded by Federal and State governments, with service providers reacting to pooled funding available through different centralised funding sources and their services not necessarily reflecting the requirements of service users. Under the NDIS, we find a demand-driven approach requiring service providers to develop and deliver programmes grounded in the demands and requirements of service users, with funding moving from direct funding of the providers to direct funding of the users through a mechanism of assessed annual support funding packages. Service users then have choice in a way not previously offered, now using service providers who can best meet their personal requirements. The landscape is changed to a more competitive and riskier marketplace, with service providers now competing for clients, against other nonprofit and for-profit organisations (Green et al. 2018). This change in focus means that service providers are now having to consider competitive market pressures, the way they structure themselves, and the skills that are now required, which are well-beyond the somewhat fewer complex parameters of service design and delivery (Rosenbaum and More 2021).

Nonprofit service providers must now grasp the new reality of both opportunities and challenges. Decision making must acknowledge the wide-ranging strategic implications that place organisational sustainability at its core, now needing to respond to service user requirements as they are indirectly funded by central governments through the NDIS funding packages paid directly to NDIS "clients" who then decide where their funding packages will be spent. This represents a shift in the approach by leadership in these organisations, where changes to organisational culture results from a change in the prevailing business models. This challenges many in this sector who have often

differentiated themselves from commercial organisations by focusing on the needs of their clients rather than, in part, on commercial outcomes that may now be required as a direct result of this shift.

This paper considers from a quantitative perspective the efficacy of the NDIS Implementation Framework ("NDISIF") (Rosenbaum and More 2022) derived from the earlier qualitative research, which identified the relevant considerations necessary for the successful implementation of the NDIS amongst nonprofit service providers. It was derived from an extensive interviewing process and considered a range of Perspectives and Influencers in a three-stage framework of Readiness → Implementation → Sustainability. Here, we report a quantitative research methodology that investigates, through surveying, the accuracy and validity of connections identified in the NDISIF. In doing so, this paper is based on a mixed methods approach to research (Denscombe 2008), with the original framework being developed and reported using a qualitative approach, whilst this follow-up paper describes a useful quantitative approach seeking to assess the validity of the key elements of the NDISIF. Both types of analyses and reporting are necessary.

As further explored in the Section 4.2 of this paper, the aim of this research is to identify the extent to which the lessons learned from the implementation of the NDIS in the nonprofit sector can be applied to the development of sector-specific change management approaches. As mentioned earlier, a mixed methods research methodology was applied in this study, combining elements of qualitative (derived through open-ended semi-structured interviews and available corporate data) and quantitative analyses (derived through detailed mathematical-based analysis of larger population-sized questionnaires). From a quantitative perspective, the detailed organisation-wide questionnaires enabled the researchers to test the Framework's validity and potentially support several research outcomes drawn from the analysis of the interview data. The use of a quantitative methodology in this manner may also enable " … generalizations to be made beyond the boundaries of the situation under study … " (Easterby-Smith et al. 1997, p. 75).

## 2. Research Question

This research seeks to validate the findings from an earlier qualitative research from which the NDIS Implementation Framework was developed. This Framework is reproduced below in Figure 1.

The elements contained within the above Framework are explained further in Table 1, which supports the key research question relevant to this current quantitative research, namely, what is the extent to which the NDISIF can be substantiated through rigorous quantitative analysis? The Conceptual Framework identified in Figure 2, including the resulting hypotheses, supports the response to this research question.

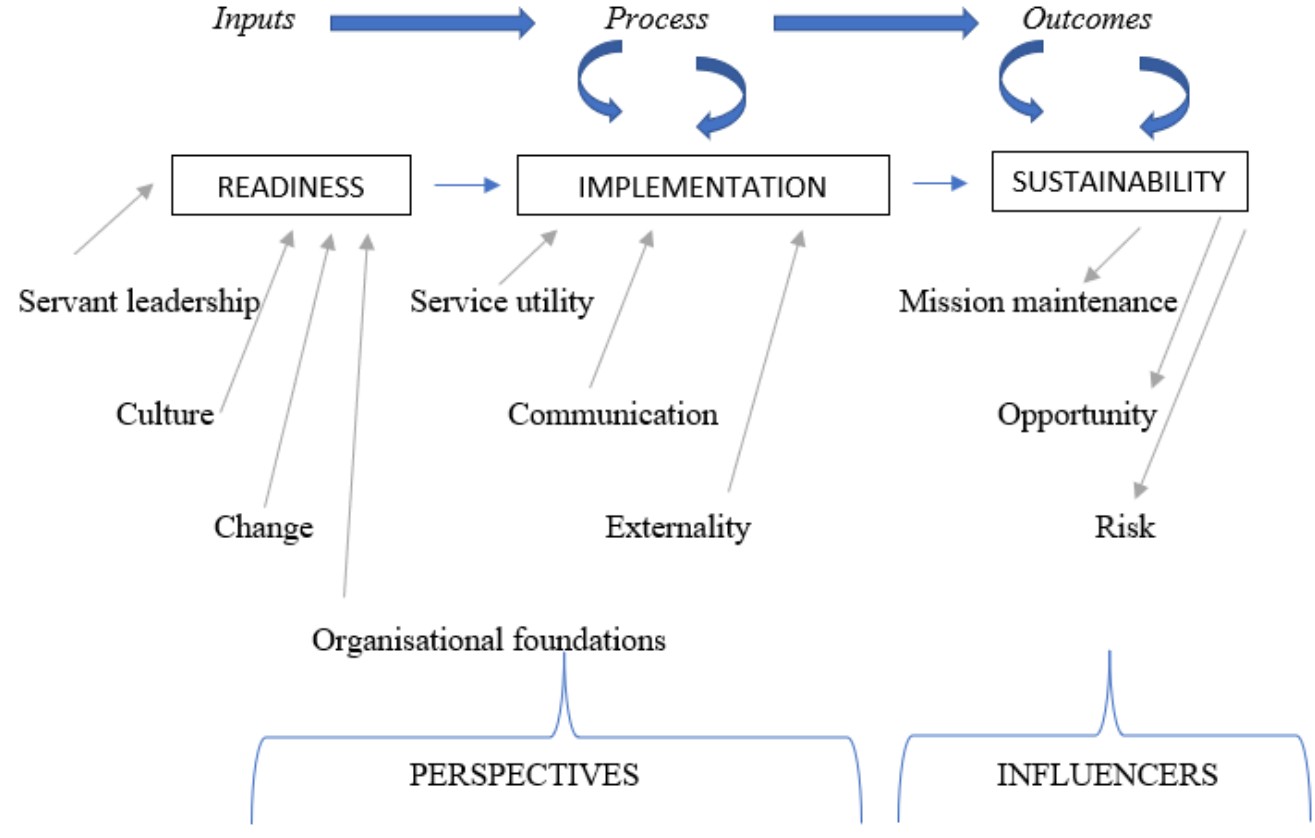

**Figure 1.** The NDISIF (Implementation Framework) (Rosenbaum and More 2022, p. 427).

**Table 1.** Success factors identified from the development of the NDIS Implementation Framework resulting from the qualitative research (Rosenbaum and More 2022).

| | **Readiness** |
|---|---|
| Leadership | Servant leadership has been identified as the most relevant leadership style for nonprofit disability service organisations implementing the NDIS. The characteristics of empowerment, humility, authenticity, interpersonal acceptance, provision of direction and stewardship, as identified in the latest research into this leadership style, support the approaches necessary in this sector (Van Dierendonck 2011). |
| Culture | Cultural adaptability (Corritore et al. 2020) and a strong focus on organisational trust (Page et al. 2019) that underpins a supportive, inclusive, empowering, and accountable culture appear as fundamental requirements in these organisations. |
| Change management | Use and application of approaches to managing change must be adaptable where the change process must be organisationally aligned and reflective of wide-ranging nonprofit attributes. An appropriate approach is the reconsidered Lewin 3-step model of change (Lewin 1947), as discussed in specific nonprofit research into change management (Rosenbaum et al. 2018). |
| Organisational foundations | The absence of a range of restricting forces, which must be addressed either before or during the change process, requires a review of the organisational structures (Waddell et al. 2019) and the role of organisational human resource functions (El-Dirani et al. 2019), and addressing issues associated with what has become known as the 'head office syndrome' (Bouquet et al. 2016). |
| | **Implementation/Commitment** |
| Service utility | The ability to provide service design and delivery in a clear manner in a contested marketplace whilst maintaining advocacy as an important element of staff engagement in a changing internal and external environment (Kimberlin 2010). |

**Table 1.** *Cont.*

| | Implementation/Commitment |
|---|---|
| Communication | Wide-ranging elements of internal communications, including coordinated top-down messaging; consistency in change communication; focused customer choice communication; addressing organisational silos linked to both service design and delivery; and the use of carefully crafted language. Additionally, communication must be authentic and sincere in order to strengthen an emotional connection and, therefore, trust between service provider and service user (Frei and Morriss 2020). |
| Externality | Reliance on effective and efficient interactions with the NDIA, which is the Federal Government Agency tasked with the rollout of the NDIS as recognised by the federal Government Joint Standing Committee on the NDIS in its 2019 Report (Andrews 2019). This reinforces the advantages resulting from a well-considered external networking approach to the implementation of the NDIS at the organsiational level. |
| | **Sustainability** |
| Mission | The ongoing maintenance of the organisational mission must be prominent in order to ensure staff acceptance of the necessary changes required to make the NDIS implementation successful (Rosenbaum et al. 2017), accepting that any apparent conflict between a values-based mission and the commercial realities of a demand-driven NDIS market place is adequately addressed from the perspective of client well-being (Dawson and Daniel 2010). This goes to the heart of organisational identity and its maintenance during all phases of the changes deemed necessary to successfully implement the NDIS (Venus et al. 2019) |
| Risk and Opportunity | The mind shift related to seeing the NDIS as an organisational and market opportunity, rather than purely a risk which requires mitigation. Such an approach supports staff in embracing the necessary changes required to successfully implement the NDIS in an uncertain and far risker context. |

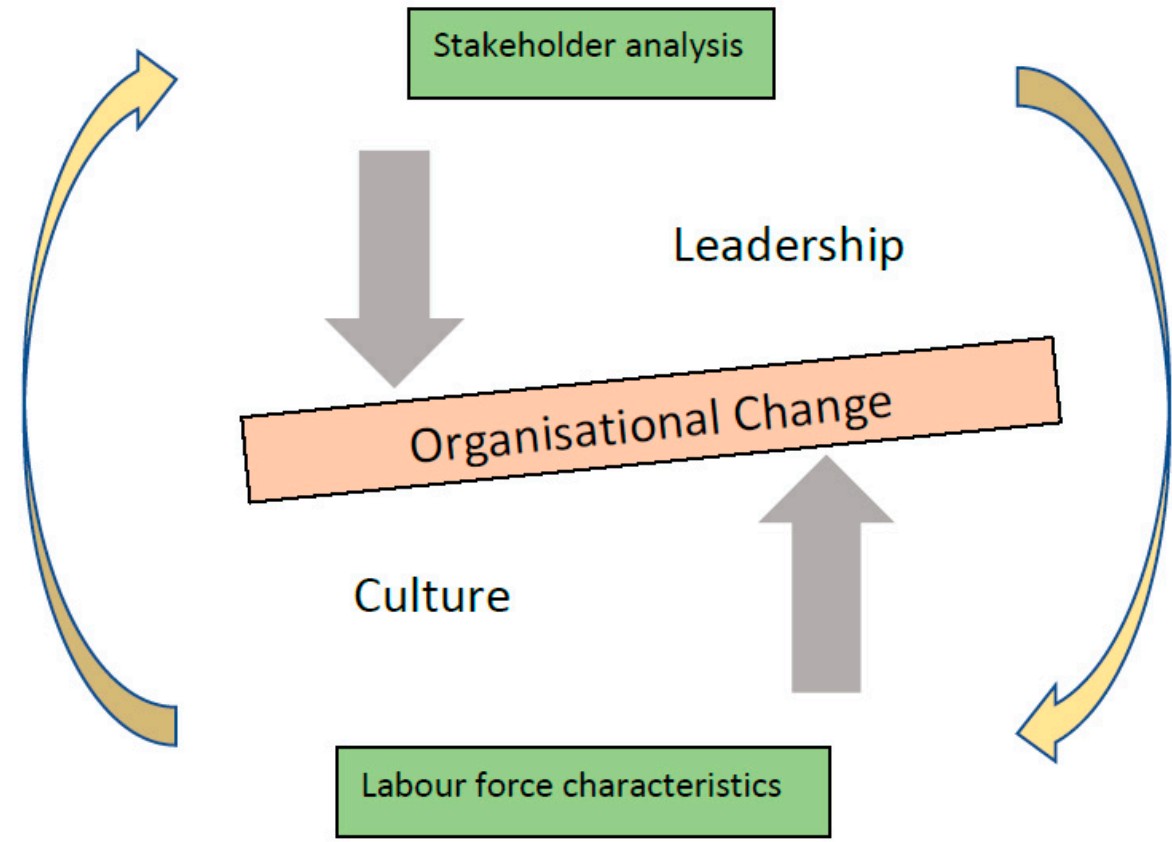

**Figure 2.** Key considerations in successful organisational change for nonprofit disability service providers.

## 3. Theoretical Context of Change

As outlined in our earlier published paper on the original qualitative aspect of our research (Rosenbaum and More 2021), the change framework for our work remains the same. Refocusing an organisation to reflect a major generational shift from its current to a future desired state though organisational change has many dimensions, including leadership styles, followership, context, nature of the change—evolutionary, change capability, culture, trust, resourcing, timing, productivity, and financial risk. Unsurprisingly, numerous models of change abound from classic to more contemporary ones (Burke 2017; Rothwell et al. 2021).

For the application of the theory, a key challenge is how to explain the way to implement organisational change that facilitates a movement from taken-for-granted mindsets, processes, and actions to new systems and being able to institutionalise such new thinking and approaches. This is exacerbated by an increasingly volatile, uncertain, complex, ambiguous, and diverse (VUCAD) environment and the need for almost constant change. This is further compounded as a direct result of our recent challenges of changes in a pandemic period that has wrought its own menu of rapid and prolonged changes.

There have been diverse approaches to understanding and explaining organisational change across numerous theoretical and disciplinary bases, from individual and group psychological and communication perspectives to those dealing with organisation-wide change. Change types also provide a different approach—incremental to radical, small- vs. large-scale change, and proactive vs. reactive. These have been well-explained some time ago by Stace and Dunphy (2001, p. 97).

Earlier approaches to change were mired in a static linear perspective, but successful change nowadays recognises the real dynamism, richness, and complexity of major change as is the NDIS, not only during implementation but also in the future to ensure the embedding of such change in the long term without individuals and organisations falling back into outmoded thoughts and practices—the unfreezing of the old state, change to the new state, and refreezing the new state for permanency (Lewin 1951).

### 3.1. Considering the Management of Change as a Sector-Specific Challenge

Organisational change management remains of substantial academic and practitioner interest (Rosenbaum et al. 2018), and, to some extent, it is supported by an ongoing dialogue as to the specifics of its execution, be that at an industry-wide level (Kätelhön et al. 2019) or at an organisation-specific level (Beniflah and Veloz 2021). From this perspective, organisational change can be viewed across a spectrum of macro-considerations through to micro-considerations. One approach sees organisational change in terms of the application of models that potentially apply to all organisations (Smith et al. 2020), thereby largely ignoring the contextual specifics that may both support and hinder successful implementation. Such an approach may, to some extent, force the processes of change through the arteries of these models, where the lifeblood of successful change develops from a focused understanding of how change unfolds. Once an understanding of this is confirmed, these models underpin an approach of adaptation. However, what may be adapting is not the model to the organisation and its context but, rather, the organisation to the model. This ignores context and reinforces a falsity that organisations in and of themselves change. However, this one-dimensional approach may prove to be inaccurate as organisations tend not to change successfully unless their people change their beliefs and adapt to the changing processes and circumstances. Therefore, the focus of change must first rest on, amongst other things, changing people's views and attitudes (Rosenbaum et al. 2018), understanding resilience (Parker and Ameen 2018), and developing employees' skill characteristics (Stouten et al. 2018) to improve change outcomes.

Following on from the potential shortcomings identified above in this one-dimensional view of change, especially with regard to the focus on individuals, is a different understanding of how change management may need to be handled. Here, we seek to include the notion of a sector-specific approach which becomes more apparent when considering

the nonprofit sector. The reason this becomes relevant is the range of characteristics that uniquely define this sector and its workforce (Rosenbaum et al. 2017), and understanding that, in any change management approach, if we fail to change the way individuals both view the change and function within it, it is likely that such change will fail, or, at the very least, result in delayed outcomes.

It is at this point that this research recognises the importance of change being considered at the level of each organisation. This suggests that unique industrial and sectoral characteristics may benefit from a framework approach to organisational change management, rather than a model approach. The latter tends to be viewed from a procedural perspective containing somewhat standardised elements, as distinct to the former approach which emphasises a wide range of contextual characteristics that must be integrated with the realities of managing change. By identifying the key Perspectives and Influencers within the broad 3-stage process of Change Readiness, Change Implementation, and Organisational Sustainability, the NDISIF provides a roadmap for how nonprofit disability service providers may implement the NDIS within their organisations (Rosenbaum and More 2022).

### 3.2. Key Considerations in Successful Change Management

From the development of the NDISIF, our research identifies a range of organisational processes and structural considerations that are necessary to maximise the change outcomes for implementing the NDIS, especially within nonprofit service delivery organisations. One key element relates to organisational flexibility, where any procedural approach to change must be balanced with appropriate leadership and cultural characteristics to ensure that the context of the setting is a key consideration. This recognises the need to create a strategic change guide that reflects the risk appetite and the variability associated with the planning and execution phases, leading to the institutionalisation of the implemented change (Rosenbaum and More 2022). Such an approach considers the comprehensive method undertaken by Lewin (Lewin 1947), which accounts for the integrated steps of action research, group dynamics, and force field analysis (Rosenbaum et al. 2018).

From a change management perspective, the involvement of stakeholder analysis appears pivotal to understanding the organisational context which enables change to be understood and structured (Vargas et al. 2019). This reflects the organisational uniqueness of the settings and further focuses attention on the human element of change—the organisational actors who need to both plan for and drive change to lead to long-term sustainability. This awareness of stakeholders also focuses attention to outside of the organisation and relates to the interactions necessary to develop and maintain the networks necessary to support change. These networks are important as they ground change in a broader context by linking not only other service providers together, but also reinforcing the necessary relationships with external government agencies that determine the necessity for change. This supports the broader issue of advocacy to further strengthen staff support during change complexities in an often highly emotional setting, given the nature of the clients these organisations deal with on a day-to-day basis.

A further element to this implementation challenge is leadership understanding of the work characteristics associated with such change. The NDIS implementations challenge the historical understanding of client service delivery. Effectively, this shifts the focus from the person who has a disability and is availing themselves of a particular service to one where a more focused commercial arrangement evolves and the person with a disability becomes the client of the service delivery organisation, and the services need to be delivered in the context of a customer relationship (Rosenbaum and More 2021). This subtle, but important, shift in the relationship, has challenged many service delivery personnel, and understanding the human side of this shift becomes an important element for consideration. Addressing this changing environment is considered fundamental to the ongoing mind shift needed to guarantee success in the implementation. The focus on this rebalancing is highlighted in Figure 2 with the fulcrum of change evidencing leadership and culture as

having essential roles in ensuring effective change outcomes. Additionally, the impact of stakeholders, both internal and external, and the skill characteristics of staff, along with the interplay between these, reflect a level of both diversity and complexity that underpin successful organisational change in this sector.

This is diagrammatically represented in Figure 2 above.

### 3.3. The Australian National Disability Insurance Scheme as the Research Setting

As identified earlier in this paper, the primary research setting is the implementation of Australia's National Disability Insurance Scheme. We have focused our analysis on several disability service providers operating in the nonprofit sector, where nonprofit service providers compete with for-profit service providers. Our interest in nonprofit service providers stems from an understanding that these organisations face the challenges of managing transformational change, as outlined above. These challenges bring into sharp focus the extra dimension of traditional mission/margin conflicts within these institutions, which are less prevalent amongst commercial providers entering the market later and not having to address the pre-implementation challenges faced by their nonprofit competitors.

### 4. Research Context: Australian Nonprofit Disability Service Sector

This research embraces a quantitative approach, motivated by a need to understand the past of nonprofit disability service delivery in this complex environment and to inform the future design and implementation aspects critical to the ongoing success of such a large generational shift in the NDIS social initiative, especially considering its escalating costs.

Our NDISIF (Rosenbaum and More 2022) is premised on the key factors of organisational readiness for change, the organisational implementation strategies that support the implementation, and the sustainability challenges (see Figure 3: Conceptual Framework) that must be addressed to make the change successful. These factors appear in Table 1 above, including the elements that support them.

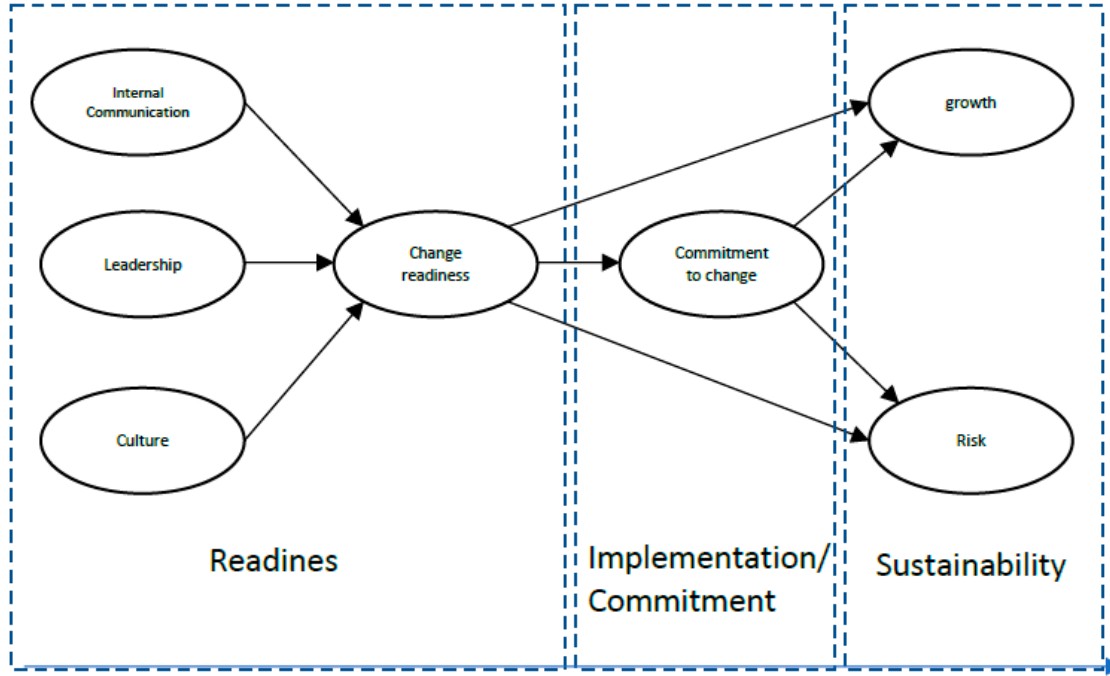

**Figure 3.** Conceptual framework developed and applied in this research stemming from the NDIS Implementation Framework developed in the earlier qualitative research, as reported in Rosenbaum and More (2022).

*4.1. Hypotheses and Associated Methodology*
4.1.1. Readiness

**H1a:** *Change readiness is significantly influenced by internal communication.*

**H1b:** *Change readiness is significantly influenced by leadership.*

**H1c:** *Change readiness is significantly influenced by culture.*

**H2a:** *Change readiness has a direct positive impact on growth.*

**H2b:** *Change readiness has a direct negative impact on risk.*

**H3a:** *Change readiness has a positive impact on growth mediated by commitment to change.*

**H3b:** *Change readiness has a negative impact on risk mediated by commitment to change.*

**H4a:** *Commitment to change has a significant mediating effect between change readiness and growth.*

**H4b:** *Commitment to change has a significant mediating effect between change readiness and risk.*

4.1.2. The Mediating Effect of Change Readiness between Change Drivers of Unfreezing and Refreezing (Sustainability)

**H1a:** *Change readiness has a significant mediating effect between internal communication and growth.*

**H1b:** *Change readiness has a significant mediating effect between leadership and growth.*

**H1c:** *Change readiness has a significant mediating effect between culture and growth.*

**H2a:** *Change readiness has a significant mediating effect between internal communication and risk.*

**H2b:** *Change readiness has a significant mediating effect between leadership and risk.*

**H2c:** *Change readiness has a significant mediating effect between culture and risk.*

4.1.3. The Mediating Effect of Commitment to Change between Readiness to Change (Unfreezing) and Sustainability (Refreezing)

    *Growth*

**H3a:** *Change readiness and commitment to change has a significant mediating effect between internal communication and growth.*

**H3b:** *Change readiness and commitment to change has a significant mediating effect between leadership and growth.*

**H3c:** *Change readiness and commitment to change has a significant mediating effect between culture and growth.*

    *Risk*

**H4a:** *Change readiness and commitment to change has a significant mediating effect between internal communication and risk.*

**H4b:** *Change readiness and commitment to change has a significant mediating effect between leadership and risk.*

**H4c:** *Change readiness and commitment to change has a significant mediating effect between culture and risk.*

### *4.2. Methodology*

Research problems and research objectives were initially explored through qualitative data collection, with the conceptual framework and the hypotheses being also developed using qualitative research and previous original research. The study reported here follows this former exploration with quantitative data that are collected from a larger sample so that the results can be better concluded for the population of interest. The purpose of collecting the quantitative data is to examine the variables with a larger sample and then perform a deeper exploration of some NDIS cases known during the qualitative phase. In particular, this quantitative study determines whether there are different mediating effects of the implementation constructs (e.g., commitment and risk) between readiness and sustainability in the NDIS Implementation Framework (NDISIF).

### 4.2.1. A Three-Stage Model Comparison

The main objective of this quantitative study is to examine whether the hypotheses regarding planned organisational change management developed in the qualitative study demonstrates differences in the drivers of change among the three stages of pre-, during, and post-implementation. This quantitative study employs a two-step analysis, namely empirical estimation and three-stage model comparison. First, this quantitative study investigates how the "moving" variables, such as "change readiness" and "commitment to change", intervene in the relationships between the unfreezing and refreezing variables. It provides findings from an empirical relationship testing of how change drivers of unfreeze factors, such as internal communication, leadership, and culture, have significant effects on the institutionalisation of organisational changes of both sustainability and risk. Second, we deal with a three-stage model comparison. The three-stage model refers to the testing of the planned organisational change management model in three different circumstances, namely pre-, during, and post-change. In this second step, a comparative analysis of the empirical testing of planned organisational change management models in three different circumstances is presented.

### 4.2.2. Sample and Procedure

This study applied partially mixed sequential qualitative—quantitative methods (Leech and Onwuegbuzie 2009). The qualitative study was conducted prior to the quantitative empirical study. The qualitative phase of the study produced a conceptual framework of planned organisational change management, as presented in the previous sections. The quantitative phase of the study recruited additional participants to those in the qualitative phase.

The questionnaire for this study was designed to expose participants to the three stages mentioned above in managing transformational change attributed to the NDIS implementation. The same set of questions was purposely designed to investigate whether there were differences in transformational changes in three different scenarios. In total, 135 employees from nonprofit service organisations completed the online questionnaire. The response rate was approximately 73%. After data cleaning and validation of excluding nonsensical responses on items, the total number of respondents who completed a set of questions pre-, during, and post-NDIS implementation totalled to 68%.

Nearly two-thirds of the sample were male (63.3%; female: 36.7%), the average age was 43.4 years (SD = 9.84), and the mean tenure was 17.85 years (SD = 11.25). Most of the sample worked in a non-managerial position (90.8%). More than half (56.2%) held a predominantly operational position, while 43.8% of the sample held a predominantly support position (administrative, IT, HR, or finance tasks supporting the operational processes).

### 4.2.3. Operationalisation Definition and Measurement

To analyse the data, this study employed partial least square (PLS) implemented in the SmartPLS program. Prior to conducting model testing, the process of turning abstract concepts of planned organisational change management into measurable observations

was developed based on the qualitative study. Table 2 shows the change driver latent variables of unfreezing as specified by the three dimensions of internal communication, leadership, and culture. The mediating variables of moving are specified by the dimensions of change readiness and commitment to change. Lastly, the refreezing variables are implied by sustainability and risk.

**Table 2.** Operational indicators of latent dimensions.

| Construct | Dimension | Definition | Indicator |
|---|---|---|---|
| Unfreezing | Internal communication | Contains elements of strong top-down but coordinated communication pathways; consistency regarding the change messaging; communication that underpins a strong understanding of the role that customer choice plays in both programme design and service delivery; recognising the need to minimise the deleterious impacts of organisational silos in the delivery of integrated services; and the effective use of wide-ranging brainstorming sessions which involve extensive cross-sections of the organisation in order to consistently address implementation challenges. Dimensions include the nature and style of language, ability to positively react to bottom-up communications, use of relevant communication media, and acknowledging authenticity and sincerity in messaging. | Intcom1: I have considered leaving this sector as a direct result of the move to a commercial model for service delivery. Intcom2: I have reluctantly shifted my approach towards the commercial realities of service delivery in the post-NDIS environment. |
| | Leadership | Focuses on the application of Servant Leadership as the appropriate focus for nonprofit disability service organisations, where leadership focuses on followers and the ability of leaders to enable followers to fully realise their own potential. | Lead1: Team-based award is a vital emotional support mechanism. Lead2: I feel that my views are considered in decision-making. |
| | Culture | Culture is represented by a range of attributes which highlight the internal organisational conflict between the purposeful mindset of service provision, based on mission, with the pragmatic reality of commercialism, as represented by the shift from a supply-driven to a demand-driven business model. This is further defined by the existence of organisational sub-cultures evidenced in internal operational silos that have historically existed within many disability service organisations. This understanding of cultural attributes is further supported by both change optimism as well as change pessimism within these organisations. | Culture1: I believe commercial practices are appropriate for nonprofit disability service providers in dealing with the NDIS implementation. Culture2: Different aspects of our organisational culture helps us to overcome difficulties. |
| | Change readiness | Change readiness identifies the extent to which the organisation can effectively introduce the necessary changes. It contains two key elements. On the one hand is the extent to which human resources within the organisation have the necessary personal skills to accept and deal with change, whilst on the other hand, it is the organisational infrastructure that supports staff as they work with the change. This latter element highlights the importance of training and upskilling staff so that the challenges of change, at the human level, can be addressed. | Ready1: People who work here feel confident that the organisation can get people invested in implementing this change. Ready2: People who work here feel confident that they can keep track of progress in implementing this change. Ready3: People who work here want to implement this change. |

**Table 2.** *Cont.*

| Construct | Dimension | Definition | Indicator |
|---|---|---|---|
| Moving | Commitment to change (Willingness to change) | Change willingness tends to be a direct product of change readiness as it is the result of the application of change skills that can then be applied to the practice of change. Willingness to change is a product of numerous factors, including an individual's understanding of change, the skills that the individual has and/or has developed to cope with change, and the attitude of the individual with regard to change. The latter can also be the product of a common vision which focuses attention on the need for change and, in the case of nonprofit disability organisations, the belief that clients of the organisation will benefit from the proposed changes. | Will1: I intend to put effort into achieving the goals of the change. Will2: I am supportive of the change. Will3: People who work here are motivated to implement this change. |
| Refreezing | Sustainability of growth | Sustainability refers to the extent to which the organisation, having undertaken the implementation and is progressively moving through the process, is able to maintain the changes over a longer period, especially when the initial change focus, which can coincide with heightened internal activity, tends to wind down. A key issue for nonprofit disability service organisations is the ability to have an ongoing focus on the original mission of the organisation to provide services to people with disabilities, whilst keeping an eye on the commercial realities that are necessary in a changing demand-driven environment. | Sustain1: I feel that my DSO has maintained its focus on its original mission and values. Sustain2: I feel that any changes to our mission and/or values are consistent with my DSO's focus on its clients. Sustain3: I agree with the need for my DSO to be 'profitable'. Sustain4: I accept the changing focus of my organisation regarding commercial outcomes that are in line with client outcomes. |
| | Risk | Risk, in the context of our nonprofit disability service organisations, is the process of understanding and managing the opportunities that invariably arise with the movement from a supply-driven to a demand-driven business model. Evaluating and mitigating risk to conceive and deal with opportunities are the foundations for ongoing sustainability by these organisations. This moves these organisations from a traditional conservative approach to one that is more reactive to options that may provide potential for growth, in terms of expanded service offerings and, therefore, financial positioning. | Riskop1: Lived experience within the organisation is considered necessary for our DSO's success. Riskop2: I believe that our DSO needs to be bold when it comes to service design in order to remain competitive. |

Unless otherwise stated, items of indicators are measured on 5-point Likert scales from 1 = *strongly disagree* to 5 = *strongly agree*.

The initial results of the qualitative study found many operational indicators to measure the dimensions of the conceptual model. After conducting a principal component analysis to reduce the items of the indicators into a set of interpretable factors for confirmatory factor analysis (CFA), an item-level analysis was used to examine the unidimensional aspects of the latent variables. The indicators of each of the latent dimensions that were used in the questionnaire are operationalised by the items, as shown in Table 2.

### 4.2.4. Strategy of Data Analysis

This study uses two steps of fit measurement evaluation, which includes the measurement model, and structural model evaluation (Chin 2010). The first step of the measurement model evaluation focuses on the validity and reliability of the indicators that are used for each latent variable, as shown in the conceptual model. The objective of the measurement model is to justify whether there are relationships between the latent variables and

its measures/indictors. Since, in this study, the conceptual model consists of reflective measurement models for all latent variables, composite reliability (rho A), convergent validity (AVE), indicator reliability (factor loading), and discriminant validity (HTMT) were evaluated (Benitez et al. 2020). The second step of structural model evaluation focuses on the assessment of the theoretical model (Akter et al. 2011). To examine the fit of the theoretical model, predictive relevance (Q2) and GoF index were used.

### 4.2.5. Assessment of the Reflective Measurement Model

The quality of the reflective measurement model was assessed using the measurement model evaluation criteria of Cronbach's alpha, rho A, average variance extracted (AVE), weight factor, and loading factor. For the first criterion, the composite reliability was assessed using Dijkstra–Henseler's rho A to estimate the correlation between the latent variable and the construct scores. The composite reliability checks the amount of random error contained in the construct scores, which is expected to be limited. The values of Dijkstra–Henseler's rho A for internal communication, change readiness, commitment to change, and sustainability as shown in Table 3, and are larger than 0.707. These values can be considered as reasonable, indicating reliable construct scores. However, the values of Dijkstra–Henseler's rho A for other latent variables, such as leadership, culture, and risk, are lower than the standard criterion of 0.707.

Convergent validity represents the extent to which the indicators' variance is explained by a latent variable. It indicates that the indicators belong to one latent variable measure, or the same construct. The average variance extracted (AVE) was used to evaluate convergent validity. Table 3 shows that all values of AVE of the latent variables are greater than 0.5, indicating that there is empirical evidence for the convergent validity of all latent variables. It means that more than half of the variance is explained by the latent variables (Bagozzi and Yi 1988).

The next assessment is indicator reliability, which measures the amount of variance presented in a latent variable in terms of the contribution of each indicator. The loadings, also called factor loadings, are a good measurement of this matter (Henseler et al. 2014). In this study, the standard estimate of factor loading is 0.707 or higher, indicating that more than 50% of the indicator variance is explained by the corresponding latent variable (Benitez et al. 2020). Table 3 shows that the factor loadings are all significant with a *p*-value less than 0.001, with estimates ranging from 0.780 to 0.919. The factor loading estimates are greater than 0.707, suggesting that the measures are reliable.

The last measurement model assessment is discriminant validity, which measures differentiation between the different aspects measured by the latent variables. It shows the degree to which a measure of construct diverges from (or has no correlation with) another measure, which underlying construct is conceptually unrelated to it. The Heterotrait-Monotrait Ratio of Correlations (HTMT) is used to provide evidence for discriminant validity. Table 4 shows that almost all the HTMT estimates are under 0.85, except $\text{HTMT}_{\text{leadership-culture}}$ and $\text{HTMT}_{\text{culture—commitment to change}}$ which are under 0.9. Discriminant validity is verified through cross loadings (Benitez et al. 2020) for pre-, during, and post-NDIS implementation, as presented in Appendix B.

**Table 3.** Measurement model evaluation.

| Construct | Indicator | Pre-NDIS Implementation | | | | During NDIS Implementation | | | | Post-NDIS Implementation | | | |
|---|---|---|---|---|---|---|---|---|---|---|---|---|---|
| | | rho A | AVE | Weight | Loading | rho A | AVE | Weight | Loading | rho A | AVE | Weight | Loading |
| Internal communication | | 0.729 | 0.76 | | | 0.737 | 0.71 | | | 0.79 | 0.761 | | |
| | Intcom1 | | | 0.567 *** | 0.888 *** | | | 0.735 *** | 0.924 *** | | | 0.688 *** | 0.927 *** |
| | Intcom2 | | | 0.561 *** | 0.886 *** | | | 0.427 *** | 0.752 *** | | | 0.445 *** | 0.814 *** |
| Leadership | | 0.665 | 0.7 | | | 0.656 | 0.719 | | | 0.646 | 0.697 | | |
| | lead1 | | | 0.591 *** | 0.872 *** | | | 0.496 *** | 0.797 *** | | | 0.469 *** | 0.759 *** |
| | lead2 | | | 0.565 *** | 0.858 *** | | | 0.675 *** | 0.896 *** | | | 0.713 *** | 0.904 *** |
| Culture | | 0.632 | 0.74 | | | 0.657 | 0.708 | | | 0.708 | 0.741 | | |
| | culture1 | | | 0.479 *** | 0.764 *** | | | 0.477 *** | 0.775 *** | | | 0.482 *** | 0.812 *** |
| | culture2 | | | 0.705 *** | 0.899 *** | | | 0.698 *** | 0.902 *** | | | 0.671 *** | 0.908 *** |
| Change readiness | | 0.897 | 0.83 | | | 0.858 | 0.771 | | | 0.898 | 0.826 | | |
| | ready1 | | | 0.357 *** | 0.897 *** | | | 0.416 *** | 0.91 *** | | | 0.375 *** | 0.923 *** |
| | ready2 | | | 0.354 *** | 0.934 *** | | | 0.354 *** | 0.885 *** | | | 0.335 *** | 0.917 *** |
| | ready3 | | | 0.389 *** | 0.897 *** | | | 0.368 *** | 0.836 *** | | | 0.391 *** | 0.886 *** |
| Commitment to change | | 0.752 | 0.78 | | | 0.79 | 0.697 | | | 0.858 | 0.779 | | |
| | will1 | | | 0.387 *** | 0.829 *** | | | 0.354 *** | 0.806 *** | | | 0.367 *** | 0.913 *** |
| | will2 | | | 0.434 *** | 0.869 *** | | | 0.407 *** | 0.874 *** | | | 0.372 *** | 0.885 *** |
| | will3 | | | 0.403 *** | 0.749 *** | | | 0.436 *** | 0.824 *** | | | 0.395 *** | 0.849 *** |
| Growth | | 0.905 | 0.78 | | | 0.906 | 0.759 | | | 0.922 | 0.777 | | |
| | sustain1 | | | 0.281 *** | 0.869 *** | | | 0.273 *** | 0.864 *** | | | 0.3 *** | 0.88 *** |
| | sustain2 | | | 0.294 *** | 0.895 *** | | | 0.324 *** | 0.911 *** | | | 0.327 *** | 0.928 *** |
| | sustain3 | | | 0.251 *** | 0.835 *** | | | 0.238 *** | 0.842 *** | | | 0.219 *** | 0.816 *** |
| | sustain4 | | | 0.313 *** | 0.905 *** | | | 0.31 *** | 0.865 *** | | | 0.282 *** | 0.899 *** |
| Risk | | 0.786 | 0.74 | | | 0.896 | 0.751 | | | 0.647 | 0.735 | | |
| | riskop1 | | | 0.458 *** | 0.831 *** | | | 0.389 *** | 0.781 *** | | | 0.549 *** | 0.84 *** |
| | riskop2 | | | 0.67 *** | 0.925 *** | | | 0.737 *** | 0.944 *** | | | 0.616 *** | 0.875 *** |

#### 4.2.6. Assessment of the Structural Model

The second step of fit measurement evaluation is the assessment of the structural model. Since this study compared the role of transformational change drivers in those three different implementation stages, the Partial Least Squares Multi-Group Analysis (PLS MGA) was employed to analyse the differences in the transformational changes. To perform PLS MGA, measurement invariance must be assessed to confirm that the measurement models specify measures of the same attribute under different conditions (Henseler et al. 2014). The measurement invariance test is meant to ensure that the construct measures are invariant across the groups (Steenkamp and Baumgartner 1998; Sarstedt and Ringle 2011). Before discussing the fit measurement testing results of the structural model, the measurement invariance was examined.

**Table 4.** Overall construct correlation matrix (HTMT).

| | Change Readiness | Culture | Internal Communication | Leadership | Risk | Growth | Commitment to Change |
|---|---|---|---|---|---|---|---|
| Change readiness | 1 | | | | | | |
| Culture | 0.742 | 1 | | | | | |
| Internal communication | 0.332 | 0.525 | 1 | | | | |
| Leadership | 0.791 | 0.89 | 0.38 | 1 | | | |
| Risk | 0.274 | 0.524 | 0.319 | 0.326 | 1 | | |
| Growth | 0.549 | 0.841 | 0.578 | 0.595 | 0.499 | 1 | |
| Commitment to change | 0.819 | 0.869 | 0.576 | 0.733 | 0.568 | 0.783 | 1 |

Note: Construct correlation matrix for pre-, during, and post-NDIS implementation is presented in Appendix B.

### 4.2.7. Test for Measurement Invariance

To assess measurement invariance, this study used the measurement invariance of composite models (MICOM) procedure developed by Henseler et al. (2014). Appendix A presents the results of three comparisons' MICOM test of compositional invariance and composite equality estimates. The results of the measurement invariance test confirmed that, generally, the multigroup comparison test results corresponded very closely since the compositional invariance and full measurement invariance were established. The justification of compositional invariance was supported by the parametric test that yielded, in all cases, higher t-values than the permutation test (all *p*-values were insignificant). It was also shown by the fact that all "original correlations" were greater than/equal to the 5% quantile. The establishment of full invariance is justified by the composite equality test, as shown in Appendix A, in which most of the mean differences and all variance differences fall between the 2.5% and 97.5% boundaries. The measurement invariance test discovers that, in respect of all three structural model relations, the three path coefficients are equal across the three stages (pre-, during, and post-NDIS implementation). Since the measurement invariance test using the MICOM procedure is achieved, the group comparisons can be proceeded with Multigroup Analysis (MGA).

### 4.2.8. Test for Structural Model

The last step of a fit measurement evaluation is a structural model assessment, which evaluates, with respect to the estimates and hypothesis tests, the causal relations between the exogenous and endogenous variables. The results of overall fit of the estimated model, such as path coefficient estimates, effect sizes ($f^2$), and coefficient of determination ($R^2$), meet the minimum model fit, as shown in Table 5. The overall fit of the estimated model was evaluated using a bootstrap-based test of the overall model fit and the SRMR. The purpose of this evaluation is to measure of approximate fit to obtain an empirical model for the proposed theory. Table 5 contains the value of the complete model SRMR, which is below the recommended threshold value of 0.080 (Henseler et al. 2014; Hu et al. 1992). However, the values of the SRMR for pre- and post-NDIS implementation are slightly greater than the threshold value due to the small sample size (N) and low degree of freedom (df) (Baron and Kenny 1986). Hu and Bentler (1999) advised that the model should be neglected if the SRMR is greater than 0.1.

**Table 5.** Structural model evaluation.

| | Path Coefficient | | | f-Square | | | |
| --- | --- | --- | --- | --- | --- | --- | --- |
| | **P1** | **P2** | **P3** | **Complete** | **P1** | **P2** | **P3** |
| Internal communication -> Change readiness | −0.092 ns | 0.208 *** | −0.007 ns | 0.18 | 0.011 | 0.089 | 0.232 |
| Leadership -> Change readiness | 0.289 ** | 0.442 *** | 0.485 *** | 0.005 | 0.089 | 0.25 | 0 |
| Culture -> Change readiness | 0.372 *** | 0.279 *** | 0.331 *** | 0.109 | 0.128 | 0.096 | 0.157 |
| Change readiness -> Commitment to change | 0.567 *** | 0.727 *** | 0.782 *** | 0.003 | 0.475 | 1.122 | 0.006 |
| Change readiness -> Growth | −0.113 ns | 0.29 *** | 0.082 ns | 0.376 | 0.017 | 0.074 | 0.36 |
| Commitment to change -> Growth | 0.754 *** | 0.444 *** | 0.662 *** | 0.941 | 0.748 | 0.174 | 1.579 |
| Change readiness -> Risk | −0.143 ns | −0.156 ns | −0.164 ns | 0.185 | 0.018 | 0.013 | 0.316 |
| Commitment to change -> Risk | 0.531 *** | 0.45 *** | 0.653 *** | 0.014 | 0.244 | 0.109 | 0.015 |
| | | | | R Square | | | |
| | | | | **Complete** | **P1** | **P2** | **P3** |
| Change readiness | | | | 0.427 | 0.308 | 0.557 | 0.507 |
| Risk | | | | 0.194 | 0.217 | 0.124 | 0.286 |
| Growth | | | | 0.452 | 0.484 | 0.468 | 0.529 |
| Commitment to change | | | | 0.485 | 0.322 | 0.529 | 0.612 |
| Overall fit of the estimated model | | | | Value | | | |
| SRMR | | | | 0.072 | 0.099 | 0.069 | 0.085 |
| d_ULS | | | | 0.883 | 1.685 | 0.824 | 1.229 |
| d_G | | | | 0.475 | 0.719 | 0.589 | 1.114 |

The second evaluation of the structural model is path coefficients and their significance levels. The path coefficient estimates for the hypothesised relationships range from −0.164 to 0.754. Most of these coefficient estimates are significant at a 5% significance level. The next structural model evaluation is to examine the effect sizes of the relationships between the constructs. This study used $f^2$ values to measure the magnitude of an effect independent of the sample size. The effect sizes range from weak to large, with the relationship between Leadership and Change Readiness having the weakest effect size. The final evaluation of the structural model is R-square which assesses goodness of fit in regression analysis. The R-square value gives the share of variance explained in a dependent construct. An evaluation of R-square values should be judged relative to studies that investigate the same dependent variable (Benitez et al. 2020). In this study, the R-square values range from 0.124 to 0.612, which are considered to be acceptable values since this study of transformational change in non-profit organisational setting is in its initial stages.

Findings

The purpose of this quantitative study is to examine empirically the moderating effects of commitment to change on the relationships between change readiness (ready to change) and organisational sustainability in three scenarios of transformational changes—pre-,

during, and post-NDIS implementation. Table 6a presents the path coefficients of the direct relationships between change readiness and sustainability, and the indirect relationships between change readiness and sustainability moderated by commitment to change. The results of the partial least squares (PLS) analysis show the significant relationships between change readiness, commitment to change, and sustainability at different levels of effects.

**Table 6.** a. Three-stage model comparisons. b. Regression coefficient comparison.

| | **a** | | | | | |
|---|---|---|---|---|---|---|
| | Pre- | | During | | Post-NDIS Implementation | |
| | Path Coeff (STDEV) | *p*-Value | Path Coeff (STDEV) | *p*-Value | Path Coeff (STDEV | *p*-Value |
| Internal communication -> **Change readiness** | −0.092 (0.097) | 0.342 | 0.208 (0.062) | 0.001 | −0.007 (0.08) | 0.927 |
| Leadership -> **Change readiness** | 0.289 (0.105) | 0.006 | 0.442 (0.071) | 0.000 | 0.485 (0.083) | 0.000 |
| Culture -> **Change readiness** | 0.372 (0.109) | 0.001 | 0.279 (0.07) | 0.000 | 0.331 (0.091) | 0.000 |
| **Change readiness** -> Risk | −0.143 (0.114) | 0.21 | −0.156 (0.126) | 0.217 | −0.164 (0.136) | 0.227 |
| **Change readiness** -> Growth | −0.113 (0.073) | 0.122 | 0.29 (0.084) | 0.001 | 0.082 (0.112) | 0.465 |
| **Change readiness** -> Commitment to change | 0.567 (0.101) | 0.000 | 0.727 (0.047) | 0.000 | 0.782 (0.046) | 0.000 |
| Commitment to change -> Risk | 0.531 (0.159) | 0.001 | 0.45 (0.119) | 0.000 | 0.653 (0.146) | 0.000 |
| Commitment to change -> Growth | 0.754 (0.062) | 0.000 | 0.444 (0.081) | 0.000 | 0.662 (0.101) | 0.000 |
| | **b** | | | | | |
| | Pre- vs. During | | During vs. Post- | | Pre- vs. Post- | |
| | Path Coeff Diff | *p*-Value | Path Coeff Diff | *p*-Value | Path Coeff Diff | *p*-Value |
| Internal communication -> Change readiness | 0.301 | 0.006 | 0.216 | 0.034 | 0.085 | 0.506 |
| Leadership -> Change readiness | 0.153 | 0.23 | −0.043 | 0.696 | 0.195 | 0.145 |
| Culture -> Change readiness | −0.093 | 0.479 | −0.052 | 0.655 | −0.041 | 0.773 |
| Change readiness -> Risk | −0.013 | 0.936 | 0.008 | 0.983 | −0.021 | 0.925 |
| Change readiness -> Growth | 0.403 | 0.001 | 0.208 | 0.142 | 0.195 | 0.144 |
| Change readiness -> Commitment to change | 0.16 | 0.114 | −0.055 | 0.39 | 0.215 | 0.027 |
| Commitment to change -> Risk | −0.082 | 0.671 | −0.203 | 0.282 | 0.122 | 0.585 |
| Commitment to change -> Growth | −0.31 | 0.003 | −0.218 | 0.095 | −0.092 | 0.435 |

### 4.3. The Effects of Change Readiness on Sustainability

Internal communication, both in terms of quantity and design, has been determined to be a key ingredient by employees in order to support organisational readiness to change during the implementation of the NDIS. The quantitative analysis supports the view that, during the implementation of the NDIS, internal communication, along with culture and leadership, supports the ability of employees to support organisational growth, leading to organisational sustainability, whilst having no effect on risk (see Table 6a). As it is presented in the difference testing of the regression coefficients, the role of internal communication is significantly different during the implementation of NDIS from the pre-NDIS implementation (see Table 6b, path coefficient difference = 0.301, *p*-value < 0.01) and post-NDIS implementation phases (path coefficient difference = 0.216, *p*-value < 0.05)). This means that internal communication plays a significant role largely during the implementation of NDIS. It can, therefore, be determined that internal communication, as described above, assists employees to accept the necessary organisational changes driving the shift from the previous supply-driven model to the newly created demand-driven model for service-provision design and delivery. In this manner, it becomes an important feature in overall change readiness considerations, both from a timing and an execution perspective.

In all three phases, employee attitude towards leadership and culture has a significant influence on the creation of organisational sustainability (growth and risk) mediated by change readiness and commitment to change.

During the implementation of the NDIS, employees must be ready to change, as reflected in Table 6a. Change readiness has both a direct and an indirect effect on the sustainability of organisational growth. Based on Table 6a, the direct effect of change readiness on organisational growth and sustainability is significant in the scenario of "during" NDIS implementation (standardised coefficient = 0.29, *p*-value < 0.001). However, change readiness does not directly affect organisational risk in relation to sustainability in the three scenarios of transformation change.

### 4.4. Commitment as a Change Moderator

In routine/regular activity, which occurs in the pre-and post-NDIS implementation phases, change readiness alone is not enough to impact organisational sustainability unless employees also have adequate commitment to the change process and recognise the need for change. Our research reflects the interdependencies associated with change readiness and change commitment. Commitment to change is needed in all situations to drive employees to make the necessary organisational changes that can lead to sustainability, both from a growth and a risk perspective (see Table 6a), recognising that the latter also reflects opportunities associated with the organisational view of a future in a post-NDIS environment. However, during the implementation of the NDIS, change commitment has a lower significant role compared to the other two phases of NDIS implementation (see Table 6a,b the regression of coefficient of implementation is lower than those of the other two phases with *p*-value < 0.001). Our research suggests that there may be higher levels of change readiness and change commitment after the implementation phase of the NDIS than before. This points to the realities of the implementation of the NDIS moving employees to a state of acceptance that could support any ongoing post-implementation changes that may be required. Accordingly, the fact that employee commitment grows as the implementation progresses reinforces the view that the pre-NDIS implementation and the implementation phases play an important role leading to the full implementation.

### 4.5. Sustainability: The Outcome of Change

Our research suggests that, in all stages, organisational sustainability (both risk and growth) is affected by the levels of change readiness and change commitment. The path coefficients between commitment to change and sustainability (both risk and growth) range from 0.44 to 0.754, with *p*-value < 0.001. However, the influence of change commitment on the growth factors of sustainability, after the implementation of the NDIS, is lower than

that before the implementation (path coefficient of 0.754 compared to 0.662), meaning a timely focus on change commitment is an important overall ingredient. Although the path coefficient difference between the pre- and post-NDIS implementation phases is not significant ($-0.92$, *p*-value = 0.435), commitment to change has a significant contribution effect on organisational growth and sustainability (see Table 5, R-squares are around 0.5).

## 5. Discussion

Continuous change in most organisations seems to be the norm, especially in healthcare, although not all are as radical a change as is the introduction and implementation of the NDIS. It appears to have proven even more complex and challenging since its first steps in 2016 and, as it continues to grow and escalate in costs, it is imperative to increase our understanding of such change and its management for success. This represents on-going challenges for organisational leadership in this sector as the NDIS reflects societal expectations which must be interpreted and managed at an organisational level, ensuring that service design and delivery can meet the expectations of those whom the NDIS was designed to service. Other challenges associated with the NDIS highlight possible linkages with socio-economic issues (Cortese et al. 2021), which, whilst focused on a policy level, will impact service delivery organisations over time as service design and delivery could, to some extent, be partially impacted by Australia's geography and population locations (Wiesel et al. 2017).

Both positive and negative views of major change management theories and models abound. Most recently, Chowthi-Williams and Davis (2022, p. 1) claim that "readiness for change could provide the energy, motivation, and engagement for successful change management. . . . " urging that leaders must deal with change management inhibitors and focus more on their people and energy." Lailla (2022, p. 404) found that "organizational change was related to changes in strategy, culture, employee attitudes, organizational structure, technology, communication leadership, and employee development affecting employee performance."

But, here, rather than taking on an a priori model and testing it, we have developed a pragmatic framework from the change management experience of a variety of nonprofit organisations dealing with the challenges wrought by implementing the NDIS. In doing so, some of the views expressed above, coming after our work, nevertheless come to the fore in our research.

Consequently, there were many lessons learnt during this research on the move from a supply-driven to a demand-driven approach, which has challenged the structures, skills, processes, and mindsets that are long-embedded and need to change in a competitive environment where client service delivery is turned on its head. The sustainability of the participating organisations emerges as a key challenge in the generational shift, leading us to explore the efficacy of the framework for successful NDIS implementation in the nonprofit sector. This is a broad change management approach in terms of its characteristics, instead of a narrow prescriptive model that would inhibit useful individual organizational characteristics and idiosyncrasies that are necessary in innovation and resilience.

Across change readiness, implementation, and sustainability, we teased out critical components for consideration, leading to the creation of the NDISIF for the sector. Stakeholder analysis, both internal and external, is important, as are other components, such as leadership and culture. Based on a qualitative study of seven organisations using grounded theory and framework analysis, for the quantitative study to consolidate and teste the findings of the original work, five more organisations were added.

The lessons learnt are in the success factors across readiness, implementation/commitment, and finally sustainability-leadership, culture, change management, organizational foundations, service utility, communication, externality, mission, risk, and opportunity. These are found to be crucial in both the qualitative and quantitative research studies.

## 6. Conclusions

### 6.1. Implications for Theory

First, the research undertaken and reported in this study adds to the body of change management work in the nonprofit sector, which is often ignored in favour of the for-profit sector. Moreover, the mixed methods approach we used is different from many other studies, ending in the creation of a novel framework, which is grounded in the initial qualitative work, through interviews primarily, and then re-examined and tested for validity in the quantitative work through surveys.

### 6.2. Implications for Practice

Given the ongoing critiques of the NDIS and the government body overseeing its operations, the National Disability Insurance Agency (NDIA), since its pilot trials in 2016, it remains important in 2022 and beyond to provide the sort of framework that our research has produced, the NDISIF, to assist the ongoing NDIS implementation challenges in the nonprofit sector. Moreover, whilst the original figure of those to be serviced by the NDIS was given as 45,000, this has now grown to be 560,000 and continues to grow, as does the ballooning cost and risks of the scheme, a challenge for whatever government is in office. Furthermore, with the current challenges of rising inflation, competition for workforce talent, and other economic and political challenges, it is imperative that research provides the much-needed support for disability organisations in the nonprofit sector to succeed, or we will fail both the provider organisations and their clients. The ramifications for nonprofit NDIS service delivery organisations are substantial. Managerial challenges for leadership in these organisations are embodied in sustainability issues stemming directly from an inability to mitigate the substantial risks associated with implementation, whilst on the other hand, failure to identify and act, in a timely fashion, on the equally substantial opportunities that present themselves as a direct result of the changes to service design and delivery that the NDIS relies on. Accordingly, the implementation of the NDIS will result in "winners" and "losers" both at the organisational level as well as amongst service users (Green and Mears 2014).

### 6.3. Future Research

We are hoping that the NDISIF framework, in providing guidance and an approach necessary, we believe, for success, can also be a useful framework for other similar social systemic change management, including both large-scale change projects as in, for example, aged care services, housing, and education, as well as other smaller-scale change projects within organisations.

We also hope to refine our methodological approach to see how useful it is in understanding some of the current "wicked problems" confronting society, especially so we can enhance it for future research in the nonprofit sector and publication in a proposed future Handbook in Nonprofit Change Management.

### 6.4. Limitations

A key limitation is that we will need to broaden the focus area of NDIS implementation as it may not always be a basis for the change management framework we have devised when the framework is tested against other change management approaches, contexts, and locations, given its characteristics may not cross over well into other challenging areas. The addition of new organisations only in the quantitative part of the research may also have had unintended consequences, although we feel this strategy enhanced the richness of the data for the study. The issue of the usefulness of the NDISIF in the for-profit sector is also yet to be explored more fully.

**Author Contributions:** All authors contributed to all sections of the paper. Additionally, All authors have read and agreed to the published version of the manuscript.

**Funding:** This research was in part funded by three industry partners within the Australian non-rpofit sector including Koorana Child and Family Services Inc. (Sydney, Australia); Junction Works Ltd. (Melbourne, Australia), and Wayside Chapel (Sydney, Australia).

**Data Availability Statement:** Data comprises responses from online survey. These are held on file by the researchers.

**Conflicts of Interest:** The authors declare no conflict of interest.

## Appendix A. Measurement Invariance of Composite Models (MICOM)

| | Compositional Invariance | | | | | | | | | Composite Equality | | | | | |
|---|---|---|---|---|---|---|---|---|---|---|---|---|---|---|---|
| | Original Correlation | Correlation Permutation Mean | 5.00% | Permutation *p*-Values | Mean—Original Difference | Mean—Permutation Mean Difference | 2.50% | 97.50% | Permutation *p*-Values | Variance—Original Difference | Variance—Permutation Mean Difference | 2.50% | 97.50% | Permutation *p*-Values |
| | | | | | | Pre vs. During | | | | | | | | |
| Change readiness | 0.999 | 0.999 | 0.997 | 0.474 | −0.362 | 0 | −0.27 | 0.267 | 0.002 | 0.191 | 0 | −0.39 | 0.387 | 0.358 |
| Culture | 1 | 0.994 | 0.978 | 0.984 | −0.054 | 0.006 | −0.249 | 0.267 | 0.699 | 0.012 | −0.003 | −0.38 | 0.368 | 0.957 |
| Internal communication | 0.989 | 0.965 | 0.873 | 0.549 | 0 | 0.001 | −0.253 | 0.253 | 0.992 | −0.081 | 0 | −0.29 | 0.272 | 0.578 |
| Leadership | 0.994 | 0.997 | 0.99 | 0.147 | −0.051 | 0.004 | −0.259 | 0.252 | 0.717 | −0.126 | 0 | −0.38 | 0.381 | 0.55 |
| Risk | 0.998 | 0.991 | 0.966 | 0.632 | −0.169 | 0.004 | −0.261 | 0.265 | 0.196 | 0.108 | −0.004 | −0.43 | 0.452 | 0.691 |
| Sustainability | 1 | 0.999 | 0.998 | 0.808 | 0.005 | 0.001 | −0.253 | 0.261 | 0.959 | −0.027 | 0.003 | −0.39 | 0.391 | 0.899 |
| Commitment to change | 0.999 | 0.999 | 0.996 | 0.532 | −0.216 | 0.008 | −0.237 | 0.253 | 0.092 | −0.102 | 0.001 | −0.34 | 0.346 | 0.555 |
| | | | | | | During vs. Post | | | | | | | | |
| Change readiness | 1 | 0.999 | 0.998 | 0.474 | 0.26 | −0.001 | −0.24 | 0.235 | 0.043 | −0.053 | 0.002 | −0.34 | 0.317 | 0.381 |
| Culture | 1 | 0.996 | 0.983 | 0.856 | −0.086 | 0.001 | −0.223 | 0.243 | 0.25 | −0.025 | 0.005 | −0.32 | 0.345 | 0.457 |
| Internal communication | 1 | 0.985 | 0.944 | 0.88 | 0.047 | 0 | −0.227 | 0.228 | 0.362 | 0.002 | −0.009 | −0.24 | 0.209 | 0.486 |
| Leadership | 0.999 | 0.997 | 0.99 | 0.64 | 0.083 | 0.001 | −0.223 | 0.226 | 0.267 | 0.05 | 0.003 | −0.32 | 0.339 | 0.408 |
| Risk | 0.99 | 0.992 | 0.968 | 0.259 | 0.182 | 0.004 | −0.22 | 0.23 | 0.096 | −0.01 | 0.003 | −0.26 | 0.283 | 0.496 |
| Sustainability | 1 | 0.999 | 0.998 | 0.516 | −0.16 | 0.002 | −0.219 | 0.237 | 0.11 | 0.11 | −0.002 | −0.36 | 0.367 | 0.305 |
| Commitment to change | 1 | 0.999 | 0.998 | 0.629 | 0.23 | 0.003 | −0.234 | 0.236 | 0.055 | −0.274 | 0.003 | −0.36 | 0.364 | 0.097 |
| | | | | | | Pre vs. Post | | | | | | | | |
| Change readiness | 1 | 1 | 0.999 | 0.826 | −0.112 | 0.003 | −0.234 | 0.234 | 0.214 | 0.142 | −0.008 | −0.32 | 0.33 | 0.224 |
| Culture | 1 | 0.994 | 0.977 | 0.878 | −0.138 | −0.002 | −0.237 | 0.225 | 0.159 | −0.013 | 0.005 | −0.27 | 0.274 | 0.456 |
| Internal communication | 0.994 | 0.951 | 0.794 | 0.679 | 0.047 | 0.002 | −0.237 | 0.238 | 0.378 | −0.078 | 0.002 | −0.25 | 0.259 | 0.283 |
| Leadership | 0.99 | 0.996 | 0.985 | 0.126 | 0.042 | 0.002 | −0.235 | 0.239 | 0.393 | −0.083 | −0.005 | −0.34 | 0.32 | 0.361 |
| Risk | 0.997 | 0.993 | 0.972 | 0.559 | 0.003 | 0.003 | −0.226 | 0.225 | 0.508 | 0.121 | −0.004 | −0.43 | 0.395 | 0.345 |
| Sustainability | 0.999 | 0.999 | 0.997 | 0.355 | −0.156 | 0.001 | −0.249 | 0.236 | 0.142 | 0.08 | 0.007 | −0.33 | 0.347 | 0.363 |
| Commitment to change | 1 | 0.999 | 0.997 | 0.66 | 0.031 | 0.002 | −0.247 | 0.229 | 0.426 | −0.373 | 0.005 | −0.35 | 0.362 | 0.032 |

**Appendix B. Construct correlation matrix of Pre-, During, and Post-NDIS Implementation**

| | Pre-NDIS Implementation | | | | | | | During NDIS Implementation | | | | | | | Post-NDIS Implementation | | | | | | |
|---|---|---|---|---|---|---|---|---|---|---|---|---|---|---|---|---|---|---|---|---|---|
| | 1 | 2 | 3 | 4 | 5 | 6 | 7 | 1 | 2 | 3 | 4 | 5 | 6 | 7 | 1 | 2 | 3 | 4 | 5 | 6 | 7 |
| 1: Change readiness | 1 | | | | | | | 1 | | | | | | | 1 | | | | | | |
| 2: Culture | 0.655 | 1 | | | | | | 0.853 | 1 | | | | | | 0.739 | 1 | | | | | |
| 3: Internal communication | 0.103 | 0.579 | 1 | | | | | 0.514 | 0.487 | 1 | | | | | 0.384 | 0.523 | 1 | | | | |
| 4: Leadership | 0.607 | 0.808 | 0.191 | 1 | | | | 0.911 | 1.055 | 0.344 | 1 | | | | 0.876 | 0.768 | 0.637 | 1 | | | |
| 5: Risk | 0.194 | 0.532 | 0.234 | 0.264 | 1 | | | 0.193 | 0.342 | 0.292 | 0.165 | 1 | | | 0.447 | 0.816 | 0.472 | 0.646 | 1 | | |
| 6: Sustainability | 0.346 | 0.843 | 0.54 | 0.5 | 0.465 | 1 | | 0.691 | 0.758 | 0.64 | 0.678 | 0.388 | 1 | | 0.656 | 0.961 | 0.551 | 0.601 | 0.743 | 1 | |
| 7: Commitment to change | 0.691 | 1 | 0.469 | 0.588 | 0.602 | 0.833 | 1 | 0.876 | 0.855 | 0.657 | 0.792 | 0.435 | 0.769 | 1 | 0.883 | 0.798 | 0.589 | 0.795 | 0.704 | 0.812 | 1 |

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
