# Peer review of "Understanding Organisational Risks and Opportunities Associated with Implementing Australia’s National Disability Insurance Scheme from the Nonprofit Service Provider Perspective—Findings from Quantitative Research"

_jrfm, doi:10.3390/jrfm15120614_

Round 1
Reviewer 1 Report
Thank you very much for the opportunity to review the article.
This paper provide useful lessons from a quantitative analysis across several nonprofit organisations undergoing generational change through implementing the Australian government’s major NDIS. The paper contributes to the field in demonstrating the usefulness of the approach in revealing how change had to occur at both micro and macro levels of the organisations involved, affecting both follower and leadership transformation, whilst simultaneously reinforcing the need to address the strategic and operational risks inherent in such transformational change.
Author Response
These are contained in the attached file

Reviewer 2 Report
The topic of the paper is quite interesting and important, but the current manuscript is poorly written in places. At some point the terminology is odd and at other points the structure of the text needs serious editing.
In addition, the manuscript is very difficult to follow, especially for those who are not familiar with the Australian National Disability Insurance Scheme Implementation Framework and papers by Rosenbaum et al. given in the reference list. Please consider explaining the background compactly early in the paper.
The empirical analysis (Structural Model) appears without clear and detailed explanation. When reporting on results describe the content of the tables in an informative manner. Right now, the tables are summarized without adding enough insight. How are the conclusions derived? Please consider explaining them properly.
The paper would be much more interesting if its exposition (titles of the sections and content of sections) and organization are improved. The introduction should also motivate and explain the empirical methodology.
The paper has several small errors, omissions or uses of odd phrases.
Page 1: Reference to NDIS
Page 2: Goal 10 and 16. Why number?
Page 4: Readiness =>Implementation => Sustainability?
Page 6-7: The focus on repositioning is highlighted in Figure 1. How?
Page 7: NFP?
Page 9: Figure 2: Ready- ?
Page 9-10: Why hypotheses are presented in section “Research context: Australian nonprofit disability services sector”?
Page 10: Title “The mediating effect of commitment to change between freezing and unfreezing growth ? and risk?
Page 10: Terms freezing and unfreezing?
page 17-18: Two Tables 3
Page 20 in table 5: Internal communication => Change readiness. What does this mean? Whole column?
Page 21: Table 5 b ?
Page 22 in table 6a and 6b: Internal communication => Change readiness. What does this mean? Whole column?
Page 25-26: Where are Appendices referred? Are they needed?
Author Response
These are contained in the watched file

Reviewer 3 Report
The authors should consider the following recommendations in order to improve the original manuscript:
- The keywords should not overlap with the title of this paper.
- To include certain relevant research questions
- The title should be revised because it is extremely long, not very suitable and does not express any link with the topics of this research journal.
- The Abstract should be more consistent with the main text of the paper, preferably structured, simple, specific, clear and unbiased.
- To extend the Theoretical Framework, by providing more relevant literature review, especially studies conducted during the last 5 years. I suggest extending the literature section by including more recent and relevant studies.
- Authors also did not provide sufficient evidence on literature review to support the hypotheses. Authors should take into consideration much more recent publications in the sphere of discussed subject matter, especially studies conducted during the last 5 years. Please discuss about Covid-19 pandemic caused by Severe Acute Respiratory Syndrome Coronavirus 2 (SARS-CoV-2) and its impact on economy. I suggest extending the literature section by including recent and relevant studies, such as for instance:
1. Batool, M., et al. (2020) How COVID-19 has shaken the sharing economy? An analysis using Google trends data, Economic Research-Ekonomska Istraživanja, DOI: 10.1080/1331677X.2020.1863830;
- To include the Questionnaire used as an Appendix.
To expand the managerial implications in the article.
- Human proofreading, English grammar and spelling correction are also required in order to improve the quality of the manuscript.
- The sources must be added under each figure and table.
- I would also like to see a well-developed discussion comparing and contrasting solution/results presented in the work with existing work and then a subsection of it presenting contributions to theory/knowledge/literature and followed by a subsection on “Implications for practice”.
- Please follow the JRFM journal instructions for authors in order to edit this research paper in correct template.
Author Response

(The authors gave the same response as above.)

Round 2
Reviewer 2 Report
The presentation is much better and the empirical analysis is well motivated. The paper is pretty interesting.